# Life expectancy drop in 2020. Estimates based on Human Mortality Database

**Stefano Mazzuco**[1]☯*, **Stefano Campostrini**[2]☯

**1** Department of Statistical Sciences, University of Padova, Padova, Italy, **2** Department of Economics, University Ca' Foscari of Venice, Venice, Italy

☯ These authors contributed equally to this work.
* stefano.mazzuco@unipd.it

**Data Availability Statement:** All data used in this manuscript are available from the Human Mortality Database www.mortality.org, Short-Term Mortality Fluctuations data is used to calculate 2019 and 2020 mortality rates. Life tables data of previous

## Abstract

In many countries of the world, COVID–19 pandemic has led to exceptional changes in mortality trends. Some studies have tried to quantify the effects of Covid-19 in terms of a reduction in life expectancy at birth in 2020. However, these estimates might need to be updated now that, in most countries, the mortality data for the whole year are available. We used data from the Human Mortality Database (HMD) Short-Term Mortality Fluctuations (STMF) data series to estimate life expectancy in 2020 for several countries. The changes estimated using these data and the appropriate methodology seem to be more pessimistic than those that have been proposed so far: life expectancy dropped in the Russia by 2.16 years, 1.85 in USA, and 1.27 in England and Wales. The differences among countries are substantial: many countries (e.g. Denmark, Island, Norway, New Zealand, South Korea) saw a rather limited drop in life expectancy or have even seen an increase in life expectancy.

## Introduction

The most striking effect of the COVID–19 pandemic is the number of deaths that most of the countries in the world have witnessed. This has also drawn the attention of researchers to the possible changes that may have occurred in mortality age patterns, as well as to the possible inequalities in the responses to the pandemic. Nevertheless, the measurement of the extent to which the COVID–19 pandemic has affected countries' mortality rates is still a strongly debated topic.

Certain researchers studied the identified COVID–19 deaths [1]; while others [2, 3] preferred to focus on excess mortality. However, the latter depends on age structure [4]. Similar to other research work [4–6], we chose to focus on changes in life expectancy. This well–known measure, quantifies the expected number of years lived by an individual of a so-called "synthetic cohort", which survival is entirely defined according to the rates of a single period [7]. Despite it refers to an hypothetical cohort, in modern times, period life expectancy is a useful indicator of mortality, available for past years, and independent from the population age structure. Note that life expectancy can be computed at different ages, but using life expectancy at birth (age 0) is the most common choice to summarize the mortality level of a population.

years also available at www.mortality.org. ASll the data are freely available upon subscription.

**Funding:** SM and SC acknowledge support from grant miur–prin 2017 project 20177BR-JXS from Ministero dell'Istruzione, dell'Università e della Ricerca.

**Competing interests:** The authors have declared that no competing interests exist.

To the best of our knowledge, most of the research that has been conducted so far has underestimated the effect of the COVID–19 pandemic–related mortality on the life expectancy figures for 2020. Several authors, for instance, have estimated that life expectancy has reduced, but, so far, this has often been concluded on the basis of incomplete information: in Spain [5], a 0.9-year decline has been predicted (updated on July 16th); in England and Wales, Aburto *et al.* [4] estimate a 0.9-year reduction for women and 1.2-year reduction for men (updated on the 47th week). In USA, Andrasfay and Goldman [6] estimated a loss of 1.13-year (updated on October 3th) based on COVID deaths.

Taking advantage of the work carried out by the Human Mortality Database team [8], who have provided the Short-Term Mortality Fluctuations (STMF) data series for several countries [9] and for both genders, we have tried to provide what we believe are more realistic estimates for life expectancy decline in 2020.

## Materials and methods

The data used for the analysis are taken from the Short-Term Mortality Fluctuations (STMF) data series for several countries [9] that have been generated by the Human Mortality Database. For each country, the STMF offers weekly death counts by age groups (0–14, 15–64, 65–74,75–85, and 85+), and the relative weekly death rates calculated suing the formula:

$$m_x^w = \frac{D_x^w}{E_x/52}$$

where, $D_x^w$ is the number of deaths in week $w$ and age group $x$, and $E_x$ is the population exposure for age group $x$. Note that population exposures are sometimes forecast by the HMD under the assumption of zero migration. When the data for all weeks are available, the annual death rate is calculated as follows:

$$m_x = \sum_{w=1}^{52} m_x^w \tag{1}$$

If the last $\tilde{w}$ weeks are missing for 2020 (this is the case for Canada) the related weekly rates are substituted with the correspondent rates for 2019, so the (1) will change as follows:

$$_{2020}m_x = \sum_{w=1}^{52-\tilde{w}} {}_{2020}m_x^w + \sum_{w=52-\tilde{w}+1}^{52} {}_{2019}m_x^w \tag{2}$$

This is not intended to serve as a forecast of the missing weekly death rates, but, instead, a "best scenario", since rates at 2019 may be considered an estimate of rates in 2020 in case there had not been excess of mortality.

There are certain countries that have a relevant number of deaths with missing weeks, and these deaths have been included using the following formula:

$$_{2020}m_x^{adj} = {}_{2020}m_x \cdot \frac{_{2020}D_x + {}_{2020}D_x^{miss}}{_{2020}D_x}$$

where, $_{2020}D_x$ is the number of deaths with non-missing weeks for the year 2020, while $_{2020}D_x^{miss}$ is the number of deaths with missing weeks for the same year. The estimation of life expectancy for the year $t$ is retrieved by constructing the related life table, starting from $_t m_x^{adj}$, following the methodology used in [10]. This methodology involves calculating the average number of person-years lived by those who have died during the interval $(x, x + n)$ $(_n a_x)$. Here, the

values of $_n a_x$ are taken from the Human Mortality Database and aggregated using the following formula:

$$_n a_x = \frac{_n L_x - n \cdot l_{x+n+1}}{_n d_x} \tag{3}$$

where, $_n L_x$ is the number of life years lived during interval $(x, x + n)$, $n$ is the length of the interval and $l_{x+n+1}$ represents the number of survivors at age $x + n + 1$. The same procedure has also been applied to years before 2020 that are available in the STMF dataset, so that the life expectancy estimated with the STMF data can be compared, as a quality check, with the life expectancy provided by the Human Mortality Database. The values are extremely close for each country (full comparison is available at https://github.com/selectPRIN/LifeExp2020_Drop/blob/main/STMF_HMD_compare.md. Not surprisingly, the aggregated weekly rates calculated using the STMF data are approximate values for the annual ones. First, the weekly data might be published based on the date of registration instead on the date of occurrence, which is the case for the HMD data, and this might assign certain deaths to a different year. Second, delayed registrations are not included in the weekly statistics even though the HMD regularly updates the STMF series, but last weeks may be incomplete; third, one year in the weekly data is not equal to a calendar year, as each year in the STMF series refers to 52 weeks. However, the comparison between the STMF-based and the HMD-based life expectancy estimates reveals that these issues do not substantially affect the results presented here.

Elaborations have been generated using R software, and the code is available at the github repository https://github.com/selectPRIN/LifeExp2020_Drop/blob/main/HMD_e0_predict.R.

## Results

The estimates of the decrease in life expectancy in 2020 are reported in Table 1 and indicate particularly high losses in several countries: the highest reduction has been registered in Russia, with a 2.16 year drop with respect to 2019. For USA the figure is 1.86, much higher than the estimates provided by [6] and even higher than what the authors refer to as "higher mortality scenario" (1.22). Such a large difference is not that surprising for two reasons: first, the above article relies on the predictions of COVID–19 deaths, and it is well known that in this way the death toll of the pandemic is underestimated [11]. Second, the predictions have been made on the basis of the models defined by the Institute for Health Metrics and Evaluation, which, similarly to many other models, are short–term forecasting tools and cannot be used for months-ahead forecasts [12]. These issues are well recognized by authors even though they hypothesize that such underestimations could have been offset by the so called harvesting effect (i.e. many deaths brought about by the COVID-19 particularly hit the frailest population, and many of them would have died during the year anyway). The estimates reported here suggest that this harvesting effect is much more limited than expected.

A particularly high life expectancy decline is also registered in some Eastern European countries (Lithuania 1.71, Bulgaria 1.59, Poland 1.42) and Spain (1.36 years), Italy (1.34) and England and Wales (1.27). The decrease in Lithuania, Bulgaria and Poland is even more significant considering that the excess of death is concentrated in the last three months of 2020 (see excess of death data in the STMF data [9]). As for Spain, the previous estimate provided in [5], based only on the first wave of the pandemic, was of 0.9-year drop, which means that the second wave has been less severe of the second one. England and Wales also show a higher drop in life expectancy than what was predicted [4]. In this case, it appears that the new B.1.1.7 variant of the virus played a significant role in contributing to a rise in mortality rates in the last weeks of 2020 [13]. A notable drop in life expectancy is found also in Chile (1.10), Belgium

**Table 1. Life expectancy at birth in 2019 and 2020 for several countries.** Elaborations from the Human Mortality Database.

| Country | $e_0$ 2019 | $e_0$ 2020 | difference |
|---|---|---|---|
| Austria | 82.32 | 81.67 | 0.65 |
| Belgium | 82.02 | 80.89 | 1.13 |
| Bulgaria | 74.47 | 72.88 | 1.59 |
| Canada | 82.66 | 82.03 | 0.63 |
| Chile | 82.21 | 81.12 | 1.10 |
| Croatia | 78.17 | 77.35 | 0.82 |
| Czech Rep. | 79.36 | 78.42 | 0.93 |
| Denmark | 81.58 | 81.96 | -0.39 |
| Estonia | 78.80 | 78.75 | 0.05 |
| England & Wales | 81.95 | 80.67 | 1.27 |
| Finland | 82.00 | 81.92 | 0.08 |
| France | 82.98 | 82.25 | 0.72 |
| Germany | 81.00 | 80.46 | 0.54 |
| Greece | 81.49 | 81.12 | 0.37 |
| Hungary | 76.21 | 75.32 | 0.89 |
| Island | 83.44 | 83.36 | 0.08 |
| Israel | 83.76 | 83.46 | 0.30 |
| Italy | 83.21 | 81.87 | 1.34 |
| Latvia | 75.04 | 74.62 | 0.42 |
| Lithuania | 75.80 | 74.09 | 1.71 |
| Luxembourg | 82.95 | 82.52 | 0.43 |
| Netherlands | 82.27 | 81.52 | 0.75 |
| New Zealand | 82.37 | 83.25 | -0.88 |
| North Ir. | 81.31 | 80.36 | 0.94 |
| Norway | 83.29 | 83.55 | -0.26 |
| Poland | 77.93 | 76.51 | 1.42 |
| Portugal | 81.65 | 80.80 | 0.85 |
| Russia | 72.52 | 70.35 | 2.16 |
| Scotland | 79.28 | 78.31 | 0.97 |
| Slovakia | 77.83 | 76.96 | 0.87 |
| Slovenia | 81.45 | 80.34 | 1.11 |
| South Korea | 84.22 | 84.37 | -0.15 |
| Spain | 83.86 | 82.50 | 1.36 |
| Sweden | 83.47 | 82.74 | 0.73 |
| Switzerland | 84.18 | 83.49 | 0.69 |
| Taiwan | 80.92 | 81.72 | -0.79 |
| USA | 79.36 | 77.31 | 1.85 |

(1.13) and Slovenia (1.11); it is also worth noting the relatively high drop in Sweden (0.73), especially when compared to neighboring countries. It should be noted that similar results, with a slighlty different methodology have been obtained by Aburto *et al.* [14].

Finally, there are several countries in which life expectancy has been affected only a little by the COVID-19 outbreak. Many of them (Norway, Denmark, New Zealand, South Korea and Taiwan) show an increase in life expectancy, while others (Finland, Estonia and Island) present a limited decrease in life expectancy with a magnitude similar to the decline registered for several countries in 2015 [15].

## Discussion

Measuring the effects of COVID-19 pandemic in terms of life expectancy drop might be useful in a comparative perspective, as other measures are flawed by issues that make comparisons more difficult: COVID-19 deaths are limited by missclassification problems [16], while excess mortality (i.e. the difference between the number of observed deaths in 2020 and the expected number that would have been observed without the pandemic) may be extremely variable, depending on the way the baseline is estimated (see Schöley [17] and Nepomuceno *et al.* [18]). One issue of life expectancy at birth is that it is often misintepreted, due to the fact it refers to an hypothetical cohort and not a real one, hence the best way to understand what the figures in Table 1 really mean, is to compare them with the life expectancy drops or gains in past years. It can be easily seen that in many countries, the drop that has been experienced on 2020 is unprecented or comparable to those occurred during the World Wars periods. Finally, it might be kept on mind that so-called "dry-tinder" effects (i.e. the increased mortality can depend on high stock of extrmely vulnerable people in the population that faced the pandemic) can affect these results, even though when it has been measured, it has been found only a modest impact of "dry-tinder" effect (see Rizzi *et al.* [19]).

## Conclusion

With the hope that the effect of the COVID–19 pandemic on mortality will be over by 2022, researchers will have to wait until the end of 2021 to fully understand its impact on populations' structures. The first estimate of the effects only for 2020 are already striking, particularly when observing an easily readable indicator such as life expectancy at birth. Unfortunately, the estimates proposed here, which to the best of our knowledge, are based on more up-to-date data than those that have been previously published, present a gloomy scenario, even more than what already presented in the existing literature. In only a year, many countries have seen a reduction of more than one year of the life expectancy at birth. The differences across countries are notable: some of them that very early blocked the spread (e.g., New Zealand or South Korea) have seen an increase in life expectancy, which is probably also due to the limitations (e.g. car use) established by the measures introduced to contain the diffusion of the virus. In contrast, in other countries, in which, as is known, the pandemic has hit a large part of the population, the results in terms of longevity have indicated a loss of more than one year in 2020 and, in some cases, even more than two years. These data offer some further information, such as the major expected differences observed between men and women in almost all countries; nevertheless, for a better understanding of how the observed losses vary across the subgroups of populations (e.g., defined by some social determinant of health), further research and data are needed.

## Acknowledgments

We gratefully acknowledge the support provided by the HMD staff, in particular by Dmitri Jdanov and Carl Boe, who gave helpful guidance in the use of the STMF data.

## Author Contributions

**Conceptualization:** Stefano Mazzuco, Stefano Campostrini.

**Formal analysis:** Stefano Mazzuco.

**Methodology:** Stefano Mazzuco.

**Supervision:** Stefano Campostrini.

**Writing – original draft:** Stefano Mazzuco, Stefano Campostrini.

**Writing – review & editing:** Stefano Mazzuco, Stefano Campostrini.

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
