## [Decision Letter · Decision Letter 0]

10 Jun 2021

PONE-D-21-15829

Life expectancy drop in 2020. Estimates from Human Mortality Database

PLOS ONE

Dear Dr. MAZZUCO,

Thank you for submitting your manuscript to PLOS ONE. After careful consideration, we feel that it has merit but does not fully meet PLOS ONE’s publication criteria as it currently stands. Therefore, we invite you to submit a revised version of the manuscript that addresses the points raised during the review process.

Specifically: Please address all comments made by reviewers .In addition ,this manuscript requires a good English edit.

We look forward to receiving your revised manuscript.

Kind regards,

Amir Radfar, MD,MPH,MSc,DHSc

Academic Editor

PLOS ONE

Journal Requirements:

Additional Editor Comments (if provided):

Reviewers' comments:

Reviewer's Responses to Questions

**Comments to the Author**

1. Is the manuscript technically sound, and do the data support the conclusions?

Reviewer #1: Yes

Reviewer #2: Yes

2. Has the statistical analysis been performed appropriately and rigorously? 

Reviewer #1: Yes

Reviewer #2: Yes

3. Have the authors made all data underlying the findings in their manuscript fully available?

Reviewer #1: Yes

Reviewer #2: Yes

4. Is the manuscript presented in an intelligible fashion and written in standard English?

Reviewer #1: No

Reviewer #2: No

5. Review Comments to the Author

Reviewer #1: This short exploration of life expectancy calculated from Human Mortality Database adds additional information and context to the impact of the COVID-19 pandemic in various countries with available data.

The concept and result are well presented, but with some errors in standard English usage that might cause some ambiguity and should be addressed.

Reviewer #2: In general this article is concise and well-written. I only have few minor corrections. In the abstract, it says the US life expectancy dropped by 1.69 years and 1.83 in Spain. These do not correspond with the data in the manuscript. Please take a look at this discrepancy and correct either the abstract or the results and table.

Minor grammatical issues:

Lines 3-5: This sentence is unclear and awkward. Please revise.

Line 79: "deacline" should be "decline"

Line 106: Replace "diffusion" with "spread"

6. PLOS authors have the option to publish the peer review history of their article (what does this mean?). If published, this will include your full peer review and any attached files.

Reviewer #1: No

Reviewer #2: No

---

## [Author Response · Author response to Decision Letter 0]

30 Jun 2021

Response to reviewers is attached.

---

## [Decision Letter · Decision Letter 1]

2 Sep 2021

PONE-D-21-15829R1

Life expectancy drop in 2020. Estimates based on Human Mortality Database

PLOS ONE

Dear Dr. MAZZUCO,

Thank you for submitting your manuscript to PLOS ONE. After careful consideration, we feel that it has merit but does not fully meet PLOS ONE’s publication criteria as it currently stands. Therefore, we invite you to submit a revised version of the manuscript that addresses the points raised during the review process.

Specifically, please address comments made by reviewer #3

We look forward to receiving your revised manuscript.

Kind regards,

Amir Radfar, MD,MPH,MSc,DHSc

Academic Editor

PLOS ONE

Journal Requirements:

Reviewers' comments:

Reviewer's Responses to Questions

**Comments to the Author**

1. If the authors have adequately addressed your comments raised in a previous round of review and you feel that this manuscript is now acceptable for publication, you may indicate that here to bypass the “Comments to the Author” section, enter your conflict of interest statement in the “Confidential to Editor” section, and submit your "Accept" recommendation.

Reviewer #3: All comments have been addressed

Reviewer #4: All comments have been addressed

2. Is the manuscript technically sound, and do the data support the conclusions?

Reviewer #3: Partly

Reviewer #4: (No Response)

3. Has the statistical analysis been performed appropriately and rigorously? 

Reviewer #3: Yes

Reviewer #4: (No Response)

4. Have the authors made all data underlying the findings in their manuscript fully available?

Reviewer #3: (No Response)

Reviewer #4: (No Response)

5. Is the manuscript presented in an intelligible fashion and written in standard English?

Reviewer #3: Yes

Reviewer #4: (No Response)

6. Review Comments to the Author

Reviewer #3: The article sounds informative and valuable. However, the following items should be considered:

• In the first line of the introduction, it is mentioned: “after one year”. It has been more than a year since the start of the pandemic. Please update.

• The exact definition of “Life expectancy at birth” should be stated in the introduction. Also, what is the difference between this factor and “life expectancy”.

• The discussion needs to cover some similar studies to compare. This part could have been written more comprehensively to better highlight and interpret the results in the context of previous research. Also, mention the limitations and strengths of the present study.

• It is better to include a separate conclusion paragraph.

• There are still some grammatical and dictation issues (e.g. line 120: WhileIn contrast, line 60: xhich is)

• The article can also be considered as a short communication, based on the editor’s decision.

Reviewer #4: (No Response)

7. PLOS authors have the option to publish the peer review history of their article (what does this mean?). If published, this will include your full peer review and any attached files.

Reviewer #3: No

Reviewer #4: No

---

## [Decision Letter · Decision Letter 2]

19 Oct 2021

PONE-D-21-15829R2Life expectancy drop in 2020. Estimates based on Human Mortality DatabasePLOS ONE

Dear Dr. MAZZUCO,

Thank you for submitting your manuscript to PLOS ONE. After careful consideration, we feel that it has merit but does not fully meet PLOS ONE’s publication criteria as it currently stands. Therefore, we invite you to submit a revised version of the manuscript that addresses the points raised during the review process.

 As mentioned by reviewer#3 ,please note in the last sentence of the abstract there are two “even”,and one of which is redundant .

We look forward to receiving your revised manuscript.

Kind regards,

Amir Radfar, MD,MPH,MSc,DHSc

Academic Editor

PLOS ONE

Journal Requirements:

Additional Editor Comments (if provided):

Reviewers' comments:

Reviewer's Responses to Questions

**Comments to the Author**

1. If the authors have adequately addressed your comments raised in a previous round of review and you feel that this manuscript is now acceptable for publication, you may indicate that here to bypass the “Comments to the Author” section, enter your conflict of interest statement in the “Confidential to Editor” section, and submit your "Accept" recommendation.

Reviewer #3: All comments have been addressed

Reviewer #4: All comments have been addressed

2. Is the manuscript technically sound, and do the data support the conclusions?

Reviewer #3: Yes

Reviewer #4: (No Response)

3. Has the statistical analysis been performed appropriately and rigorously? 

Reviewer #3: Yes

Reviewer #4: (No Response)

4. Have the authors made all data underlying the findings in their manuscript fully available?

Reviewer #3: Yes

Reviewer #4: (No Response)

5. Is the manuscript presented in an intelligible fashion and written in standard English?

Reviewer #3: Yes

Reviewer #4: (No Response)

6. Review Comments to the Author

Reviewer #3: Dear authors,

Thank you for considering the comments.

In the last sentence of the abstract there are two “even”s, one of which is extra. Please do not forget to omit it.

Reviewer #4: (No Response)

7. PLOS authors have the option to publish the peer review history of their article (what does this mean?). If published, this will include your full peer review and any attached files.

Reviewer #3: No

Reviewer #4: No

---

## [Author Response · Author response to Decision Letter 2]

19 Oct 2021

"In the last sentence of the abstract there are two “even”s, one of which is extra. Please do not forget to omit it."

We dropped the duplicate "even" in the abstract

---

## [Decision Letter · Decision Letter 3]

7 Jan 2022

Life expectancy drop in 2020. Estimates based on Human Mortality Database

PONE-D-21-15829R3

Dear Dr. MAZZUCO,

We’re pleased to inform you that your manuscript has been judged scientifically suitable for publication and will be formally accepted for publication once it meets all outstanding technical requirements.

Kind regards,

Amir Radfar, MD,MPH,MSc,DHSc

Academic Editor

PLOS ONE

Additional Editor Comments (optional):

Reviewers' comments:

Reviewer's Responses to Questions

**Comments to the Author**

1. If the authors have adequately addressed your comments raised in a previous round of review and you feel that this manuscript is now acceptable for publication, you may indicate that here to bypass the “Comments to the Author” section, enter your conflict of interest statement in the “Confidential to Editor” section, and submit your "Accept" recommendation.

Reviewer #3: All comments have been addressed

Reviewer #5: All comments have been addressed

2. Is the manuscript technically sound, and do the data support the conclusions?

Reviewer #3: Yes

Reviewer #5: Yes

3. Has the statistical analysis been performed appropriately and rigorously? 

Reviewer #3: Yes

Reviewer #5: Yes

4. Have the authors made all data underlying the findings in their manuscript fully available?

Reviewer #3: Yes

Reviewer #5: Yes

5. Is the manuscript presented in an intelligible fashion and written in standard English?

Reviewer #3: Yes

Reviewer #5: Yes

6. Review Comments to the Author

Reviewer #3: Thanks for revision

It is now acceptable for publication in Plos one……………………………………………………………………………..

Reviewer #5: It is OK. I think all comments have been addressed. This manuscript is very well. I have not comments.

7. PLOS authors have the option to publish the peer review history of their article (what does this mean?). If published, this will include your full peer review and any attached files.

Reviewer #3: **Yes: **Mitra Amini

Reviewer #5: **Yes: **Masoud Behzadifar

---

## [Editor Report · Acceptance letter]

21 Jan 2022

PONE-D-21-15829R3 

Life expectancy drop in 2020. Estimates based on Human Mortality Database 

Dear Dr. Mazzuco:

I'm pleased to inform you that your manuscript has been deemed suitable for publication in PLOS ONE. Congratulations! Your manuscript is now with our production department. 

Kind regards, 

on behalf of

Dr. Amir Radfar 

Academic Editor

PLOS ONE